# Tipping the balance: A systematic review and meta-ethnography to unfold the complexity of surgical antimicrobial prescribing behavior in hospital settings

Hazel Parker[1]*, Julia Frost[2], Jo Day[2], Rob Bethune[3], Anu Kajamaa[4], Kieran Hand[5], Sophie Robinson[2], Karen Mattick[2]

1 Pharmacy Department, Royal Devon University Healthcare NHS Foundation Trust, Exeter, United Kingdom, 2 College of Medicine and Health, University of Exeter, Exeter, United Kingdom, 3 Royal Devon University Healthcare NHS Foundation Trust, Exeter, United Kingdom, 4 Faculty of Education, University of Oulu, Oulu, Finland, 5 Medical Directorate, NHS England & NHS Improvement, London, United Kingdom

* hp446@exeter.ac.uk

**Data Availability Statement:** All relevant data are within the paper and its Supporting Information files.

## Abstract

Surgical specialties account for a high proportion of antimicrobial use in hospitals, and misuse has been widely reported resulting in unnecessary patient harm and antimicrobial resistance. We aimed to synthesize qualitative studies on surgical antimicrobial prescribing behavior, in hospital settings, to explain how and why contextual factors act and interact to influence practice. Stakeholder engagement was integrated throughout to ensure consideration of varying interpretive repertoires and that the findings were clinically meaningful. The meta-ethnography followed the seven phases outlined by Noblit and Hare. Eight databases were systematically searched without date restrictions. Supplementary searches were performed including forwards and backwards citation chasing and contacting first authors of included papers to highlight further work. Following screening, 14 papers were included in the meta-ethnography. Repeated reading of this work enabled identification of 48 concepts and subsequently eight overarching concepts: hierarchy; fear drives action; deprioritized; convention trumps evidence; complex judgments; discontinuity of care; team dynamics; and practice environment. The overarching concepts interacted to varying degrees but there was no consensus among stakeholders regarding an order of importance. Further abstraction of the overarching concepts led to the development of a conceptual model and a line-of-argument synthesis, which posits that social and structural mediators influence individual complex antimicrobial judgements and currently skew practice towards increased and unnecessary antimicrobial use. Crucially, our model provides insights into how we might 'tip the balance' towards more evidence-based antimicrobial use. Currently, healthcare workers deploy antimicrobials across the surgical pathway as a safety net to allay fears, reduce uncertainty and risk, and to mitigate against personal blame. Our synthesis indicates that prescribing is unlikely to change until the social and structural mediators driving practice are addressed. Furthermore, it suggests that research specifically exploring the context for effective and sustainable quality improvement stewardship initiatives in surgery is now urgent.

**Funding:** This report is independent research supported by Health Education England and the National Institute for Health Research (HEE/ NIHR ICA Program Clinical Doctoral Research Fellowship, Miss Hazel Parker, ICA-CDRF-2018-04-ST2-019) and the National Institute for Health and Care Research Applied Research Collaboration South West Peninsula. The views expressed in this publication are those of the authors and not necessarily those of the NHS, the National Institute for Health Research or the Department of Health and Social Care. The funder, sponsor and authors' institutions have had no role in developing the protocol.

**Competing interests:** The authors have declared that no competing interests exist.

# Introduction

Antimicrobial resistance (AMR) poses a significant threat to patient safety, globally. Unless policies are successfully implemented to tackle its spread, by 2050, it is estimated that it will be responsible for ten million deaths annually [1]. There is a direct correlation between antimicrobial consumption and resistance [2] so antimicrobial stewardship, encompassing the careful and responsible use of antimicrobials, is one of the key strategies, alongside infection prevention and control, to reduce AMR. Stewardship programs seek to optimize clinical outcomes and to prevent antimicrobial misuse–namely unnecessary use; incorrectly timed or excessively prolonged use; and administration without due consideration of pharmacokinetic and pharmacodynamic principles–which can lead to increased AMR; hospital acquired infections; and other antimicrobial-associated adverse events. For example, 20% of hospitalized patients prescribed an antimicrobial will experience at least one antibiotic-associated adverse event [3]. However, antimicrobial prescribing is uniquely challenging. Clinicians (who may not be experts in infection management) are often required to make complex judgments with incomplete information; and the process can be both unpredictable and inconsistent [4]. They must balance the threat of AMR against that of potential sepsis, which is often framed as a more concrete threat [5]. Many believe the patient's immediate risk outweighs the long-term disadvantages of prescribing an antimicrobial [6–8] resulting in overuse and the consequences therein.

In England, hospital antimicrobial consumption (which accounts for 21% of total human consumption) has increased 8.8% over the past five years (or 3.5% if adjusted for increased admission rates) despite various improvement efforts [9] such as the wider availability of antimicrobial formularies and guidelines, and increased numbers of specialist antimicrobial pharmacists [10]. Surgical specialties account for a high proportion of antimicrobial prescribing throughout Europe [11]; and inappropriate use has been widely reported, in England [12,13] and elsewhere [14–17], with numerous calls for action [18]. A recent prospective cohort study [19], comparing antimicrobial prescribing between general surgery and general medicine, found that in surgery antimicrobials are prescribed more frequently and for longer durations; they are also more likely to be escalated (to broader-spectrum agents which are active against a wider range of bacteria) and less likely to be compliant with local guidelines. Surgeons are also reported to be less likely to consult antimicrobial prescribing guidelines than their medical counterparts [20].

Large numbers of patients (around 10 million) undergo surgery within the National Health Service (NHS) in the United Kingdom each year [21] and advances in surgical technique and anesthesia mean that growing numbers of patients, including those at increased risk of infection, can be offered surgery [22]. On any given day, 39.5% of surgical patients in English acute-care hospitals are prescribed an antimicrobial [12]. This might be a single dose of surgical antimicrobial prophylaxis (SAP), which is vital for many procedures to limit surgical site infection (SSI) [22]. However, in English hospitals, of the 1 in 12 patients administered SAP: about half receive more than the recommended single dose and a third receive more than 24 hours of antimicrobial cover [12]. The story is similar across Europe where roughly one in seven antimicrobial prescriptions are for SAP (representing the third most common indication for antimicrobials), and over half of SAP prescriptions have a duration of more than a day [11]. This is despite evidence from randomized controlled trials indicating that, for the majority of procedures, prolonged postoperative SAP has no benefit in reducing SSI after surgery when compared to a single dose [23] but does increase the risk of adverse effects. Antimicrobial treatment is only indicated for patients that develop infection, for example, a SSI or pneumonia. However, surgical patients are at increased risk of infection and the prevalence of

healthcare-associated infections is high (8.5% in the UK; 6.7% across Europe), second only to intensive care patients [12,24].

Antimicrobial misuse, within surgical specialties, has been attributed to a variety of factors including a lack of training, experience or confidence; inadequate knowledge of local AMR epidemiology; misinterpretation of microbiology results; uncertain diagnosis and/or lack of guidance or institutional leadership [25]. More recently, qualitative studies have begun to provide deeper insights into surgical teams' antimicrobial prescribing behavior (APB) highlighting it as distinct from other physicians' APB [26–28]. However, the most effective way to improve practice remains unclear and much of the previous research in this area, including some of the qualitative work, has been atheoretical.

The wide availability of guidelines indicates that providing guidance is insufficient to change practice [29]; and interventions tailored to prospectively-identified barriers are more likely to improve the situation [30]. However, currently, most interventions that are implemented to improve antimicrobial prescribing for hospital inpatients do not use the most effective behavior change techniques [31] and are not theoretically informed [32]. Interventions that are disjointed from the reality of how prescribing decisions are made are less likely to be effective [33]. Yet, little attention has been paid to how interventions work, in different contexts and for different prescribing groups, with the emphasis frequently placed on knowledge and skills [34] which are often not the only issue. The assumption of a linear knowledge to practice relationship is unhelpful, and unsuited given the complexity involved [35]. To bring about meaningful, sustained behavior change it is essential to address the cultural and contextual factors that underpin APB [36,37]. This means incorporating an understanding of the contextual factors affecting APB, for example prescribing norms and hierarchical power dynamics, into the design and delivery of context-specific, sustainable interventions. It is essential to understand what will work and for whom; however, this approach is lacking in most stewardship interventions [31]. There is an urgent need to improve surgical APB, to minimize unintended patient harm, but it is a complex and poorly understood issue.

Qualitative evidence synthesis is a mechanism to bring together isolated qualitative studies to provide accumulated understanding of phenomena–delivering fresh insights, identifying research gaps, and facilitating conceptual development and theory building [38]. A synthesis of qualitative studies, describing surgical APB, is needed to underpin the development of transferable theory to inform future research and antimicrobial stewardship programs; and to reduce avoidable research waste (e.g. researching what is already known) and inefficiency [39]. Meta-ethnography is a well-established qualitative synthesis methodology that has been widely adopted in health and social care research [40,41]–including to successfully develop theory about antimicrobial prescribing interventions in general practice [42]; and to explain the pressures and dilemmas influencing APB (across specialties) in acute hospitals in developed healthcare systems [43]. Our aim is to synthesize qualitative studies on surgical APB (which has been highlighted as distinct from other clinicians' APB) to explain how and why contextual factors act and interact to influence APB amongst surgical teams in hospital settings. Working with key stakeholders, we will develop new clinically applicable theory, which will advance understanding to inform the development of interventions and identify knowledge gaps to inform further research.

## Methods

The meta-ethnography was registered with the International Prospective Register of Systematic Reviews (PROSPERO), registration number CRD42020184343; and the published protocol [44] follows the seven meta-ethnography phases outlined by Nobit and Hare (1988). Some

**Table 1. Key terms.**

| Term | Definition |
|---|---|
| Authors | Authors of the primary studies included in the meta-ethnography |
| Reviewers | The researchers conducting this meta-ethnography (who reviewed the primary studies) |
| First order data | Primary study participant interpretations e.g. quotes from study participants |
| Second order data | Primary study author interpretations (of participant interpretations) e.g. quotes from the primary study's discussion section |
| Third order data | Reviewer interpretations based on analysis of the first order and second order data (interpretations of interpretations of interpretations) i.e. the overarching concepts developed using translation; and the conceptual model (visual representation of the line-of-argument synthesis) |
| Concepts | Explanatory ideas that have some analytic or conceptual power (unlike more descriptive themes, which characterize the essence of the data). Concepts provide an explanation of function or potential–they should be explained within the primary papers and substantiated by first order data |
| Line-of-argument synthesis | Primary studies identify different aspects of a larger phenomenon which when taken together offer a new interpretation; a 'whole' is discovered from a set of parts |

small innovations from the protocol were made (outlined below), e.g. minor amendments to the screening criteria to assess papers for inclusion in the study (Table 2); and the decision against conducting a CERQual assessment to evaluate the review findings. This decision was taken as the extensive stakeholder engagement provides confidence in the legitimacy of the review findings and fits better with the ethos of our ongoing research endeavor. Furthermore, there is no cohesive answer as to who should perform CERQual [45] and limited experience of applying it to the overarching concepts produced via meta-ethnography [46]. Although presented linearly in this manuscript, the phases of the meta-ethnography inevitably overlapped. Key methodological terms are highlighted in Table 1. Ethical approval was not required as primary data was not collected (http://www.hra-decisiontools.org.uk/ethics/).

**Table 2. SPIDER table of study inclusion and exclusion criteria [amended from Parker, Frost (44)]\*.**

| | Inclusion criteria | Exclusion criteria |
|---|---|---|
| Sample | • Surgical teams (any members including surgeons, trainee surgeons, anesthetists, surgical nurses, surgical pharmacists etc.)<br>• Secondary care setting including wards; out-patient clinics; theatres etc. | • Non-surgical specialties<br>• **Mixed specialty participants where surgical participants do not clearly make up over 50% of the sample**<br>• Other care settings e.g. primary care; dentists<br>• Veterinary studies |
| Phenomenon of interest | • Antimicrobial/antibiotic prescribing behavior (treatment and/or prophylaxis) | • Prescribing behavior related to other medication classes |
| Design | • Qualitative or mixed method studies reporting primary qualitative data collected using qualitative methods (e.g. through direct observation; focus groups; or interviews) | • Studies that report quantitative data only including questionnaire studies with open-ended free text questions |
| Evaluation | • Qualitative analysis of antimicrobial prescribing behavior (using any qualitative evaluation e.g. grounded theory; and framework analysis) | • Studies that evaluate using quantitative methods only<br>• Studies that do not explicitly state the method of analysis |
| Research type | • Peer-reviewed journal articles<br>• Full text available<br>• English language | • Reviews; protocols; theoretical work; editorials; opinion pieces; and grey literature<br>• Non-English language |

\*The amendment is shown in bold.

## Theoretical perspective

Meta-ethnography is an inductive, highly interpretive approach, which is well suited to the development of conceptual insights [38,47]. We adopted a socio-cultural-historical perspective–acknowledging that surgical APB is likely influenced by a range of factors including organizational context and workplace relationships–and sought to develop new interpretations, from first and second order data, from relevant primary studies. The research team members have a range of interpretive repertoires. HP is an experienced hospital pharmacist, specializing in the field of antimicrobials, and a National Institute for Health Research (NIHR) Clinical Doctoral Research Fellow; RB is a Consultant Colorectal Surgeon with a special interest in patient safety, human factors, and quality improvement; KH is an experienced hospital specialist microbiology pharmacist and leads the Antimicrobial Prescribing Optimization work stream of the NHS England AMR Program; JF is a Medical Sociologist and was previously a Registered Nurse; JD is a Psychologist and qualitative researcher in implementation science; AK is an educationalist and expert in sociocultural and practice-based theories, including Cultural Historical Activity Theory (CHAT); and KM is a Professor of Medical Education (also see author information). To further broaden perspective and to support the development of theory [41], we also approached a variety of healthcare workers for clinical insight and experiential knowledge at various stages throughout the meta-ethnography and worked with two stakeholder groups: (1) a group of healthcare workers with relevant surgical experience; and (2) a group of patients with first-hand experience of surgical care. Healthcare workers (n = 8) were recruited via the lead author's professional networks and included a Consultant Surgeon, Surgical Trainee, Clinical Academic GP (with experience of working in surgical rotations), Consultant Anesthetist, Trainee Anesthetist, Consultant in Microbiology and Infection, Specialist Infection Pharmacist, and a Specialist Surgical Pharmacist. Patient representatives (n = 6) were recruited via the Peninsula Public Engagement Group (PenPEG) which is a group of patients, service users and carers who volunteer their time to help ensure that research is relevant to the needs of the community (see https://arc-swp.nihr.ac.uk/ppie/). Stakeholders contributed ideas and gave feedback throughout the research process, from design to delivery, via formal 90 minute stakeholder meetings (n = 4) and at numerous informal one-to-one meetings (which varied in length from 30 minutes to half a day).

## Formulating the research question

Our research question is: how and why do contextual factors act and interact to influence surgical antimicrobial prescribing in hospital settings? Meta-ethnography was selected as the most suitable approach because it is systematic; has the potential to preserve interpretive properties from the primary studies; and can facilitate the development of conceptual understanding [47] thus moving the field forward [48]. This matters because much of the previous research in this area has not developed cumulative insights, and has not therefore developed our theoretical understanding of the problem and potential solutions. As far as we know, this research question has not been explored previously using meta-ethnography.

## Data sources and search strategy

We conducted a systematic search focusing on qualitative studies that explored APB among surgical teams [44]. The SPIDER tool [49] provided structure and clarity for the search. Search terms were individualized for each database; and refined following scoping searches to ensure that ten relevant primary papers (already known to the reviewers) were identified. In June 2020, eight databases were systematically searched, from inception, including Medline, Medline in Process, Embase, Cochrane, PsycInfo, AMED, CINAH and Web of Science (see

supporting information for the full searches). Additionally, because qualitative literature can be hard to find [50], we also: (1) conducted forwards and backwards citation chasing from the 29 papers that were initially thought to be suitable for inclusion, using Scopus (Elsevier); and (2) wrote to first/corresponding authors of the 14 papers that were ultimately included in the meta-ethnography, asking them to identify any additional relevant papers. The database search was updated in May 2021 to ensure that we captured any recently published eligible studies. All search results were exported into Endnote X9 (Thomson Reuters, NY, USA) and de-duplicated using automatic and manual checking.

### Inclusion and exclusion decisions

We used the screening criteria defined previously in our protocol [44]. However, we made an additional pragmatic decision to exclude studies with ≤50% surgical participants to ensure the findings were representative of surgical teams (see Table 2). One reviewer (HP) screened all titles and abstracts identified via the searches, with a second reviewer (SR) screening over 90%. The few discrepancies (<1.5%) between reviewers were re-read, discussed, and either identified as clearly excluded or added to the titles for full text review. Again, one reviewer (HP) reviewed all full text articles; and three reviewers (KM, JF and JD) reviewed a subset (>50%) leading to a unanimous decision to put forward 14 papers for inclusion in the meta-ethnography.

Quality assessment, using the CASP Qualitative Checklist [51], was used to support careful reading of the studies but not to exclude studies as lower scores do not always correlate with research quality (due to e.g. abridged methodological reporting); or indicate where the rich elaborations, that are much needed for meta-ethnography, will be found. Papers presenting mainly descriptive data are likely to offer fewer insights, while those that include thick descriptions and rigorous analysis will contribute more substantively [52]. As such, the inclusion of weaker papers (e.g. those that are conceptually limited) is unlikely to distort the synthesis [38]. However, each of the included papers were also classified as a key paper, or otherwise (satisfactory and questionable papers), based on their perceived utility to the meta-ethnography using a pragmatic approach first described by Dixon-Woods, Sutton [53] [also see Parker, Frost (44)]. This enabled us to examine the synthesis messages derived from the included studies against 'key' papers (only) to test their contributions and to promote further discussion and insight, consistent with previous work [54].

### Identifying concepts and how the studies are related

Included studies were repeatedly read by one reviewer (HP) to identify concepts from any section of the paper that were relevant to the research question. Some papers were already known to the reviewers and, as the process of repeated reading progressed, different papers were examined in different sequences, e.g. by author/research group, to enable initial identification of potential concepts and the nature of the relationships between them. Once the first reviewer (HP) was satisfied that all potential concepts were identified, three reviewers (KM, JF and JD) independently reviewed a subset of the included papers so that all papers were assessed by at least two reviewers. Concepts were identified and defined in a word document, substantiated by first and second order data from the primary papers that were coded using NVivo 12 (QRS International). Wherever possible, as a way of remaining faithful to the meanings and concepts in the primary studies, text and terminology were retained [55]. Additionally, study characteristics were recorded in a Microsoft Excel table to provide context for the interpretations.

Next, we began to explore the relationship across studies and between concepts. We did not group studies by focus (e.g. by surgical pathway stage) because it would have detracted from the holistic picture provided by the dataset and may have impinged upon our ability to capture

some of the important features. Furthermore, the relatively small number of papers (n = 14), and focused review question, made it possible to manage the dataset as a whole. Working through the studies chronologically, contextual data and the study concepts (with their definition) were entered into an excel spreadsheet. Each new concept was added sequentially to allow for comparison (and where concepts appeared to explain a common issue they were entered into the same column). The overarching concept/column categories were then double checked by examining each contributing concept in isolation (with reference to the primary paper as necessary) and interrogating its affiliation until we were satisfied that each concept was situated in the correct column (overarching concept).

## Translating studies into one another

The goal of translation is to produce overarching concepts (third order data), which involves interpretive reading of meaning but not further conceptual development [54], using a process that has been likened to the constant comparative method used in grounded theory [41]. As the translation proceeded, concepts and emerging explanations were revisited and compared leading to the merging of some overarching concepts if they were not clearly distinct; and to the development of clearer, more fitting conceptual titles. To craft our summary definition (reciprocal translation) for each of the overarching concepts, we looked at each primary paper's concept (affiliated to the given overarching concept) and developed a narrative that had meaning for all of the contributing papers. This was undertaken with reference to the primary papers, and with due consideration of the boundaries of other emerging overarching concept definitions (translations). Where possible, we used the primary authors' own words, or paraphrased several papers to summarize in the most meaningful way, with the intention of preserving the meaning in the primary studies. Although we actively sought disconfirming or contradictory findings, we did not identify any concepts that strongly opposed one another and as such could not produce any refutational translations.

## Synthesizing the translations

Meta-ethnography can progress from the reciprocal translations to a higher order interpretation (line-of-argument synthesis) that distils translations into more than the parts alone imply, a 'whole' is discovered from the set of parts [47,52]. However, the worth of the synthesis is to be determined by the quality of its concepts, and whether the intended audience regards the synthesis as useful [47]. Initial translations were shared with stakeholders (healthcare workers and patient representatives) to ensure that the overarching concepts and language were clear; and to explore how the overarching concepts interacted, and the extent to which they resonated with experience. This informed the development of several potential conceptual models that could explain surgical APB, which were shared with subsets of the stakeholders. Further discussion between reviewers and with stakeholders identified a lead model (visual representation of the line-of-argument synthesis) which was iteratively developed with a Surgical Consultant (RB) during a half day meeting (02/09/21). The resultant model was then further refined with input from the wider group of stakeholders, both at formal stakeholder meetings and via informal discussion, in-keeping with Spicer's call for an emic, holistic, historical and comparative approach [47].

## Results

### Included papers

The initial database search produced 5824 abstracts; the update search (May 2021) produced a further 577; and forwards and backwards citation chasing from 29 papers produced a further

1510. After deduplication, 4921 titles and abstracts remained; and following title and abstract screening 151 papers were put forward for full text review for eligibility. Full texts were excluded because the sample did not clearly consist of over 50% surgical team members (n = 70); the phenomenon of interest was not well enough aligned with our research question (n = 45); the study design was not qualitative (n = 6); the method of evaluation was not clear (n = 2); the full text was unavailable (n = 1); and the studies were duplicates that were not caught by our deduplication process (n = 13). This left 14 papers, from ten different authors, which met our inclusion criteria and were included in the meta-ethnography (see Fig 1) [26–28,56–66]. No additional suitable papers were identified via the authors of our included papers (although four of the nine corresponding authors did respond). Several papers had shared or overlapping datasets (Broom, Broom (28) and Broom, Broom (57); Charani, Tarrant (27) and Charani, Ahmad (26); and Bonaconsa, Mbamalu (64) and Singh, Mendelson (63), thus the dataset represented 11 separate bodies of work. The included research was conducted in a range of countries including: Australia (5); England (1); United States of America (2); Italy (1); India (1); and South Africa (1). All the studies were published, in English, between 2016 and 2021. A range of qualitative techniques were used to gather data including: focus groups and/or interviews (7 bodies of work/8 papers); and ethnographic methods including observation, interviews, and in one case illustration of sociograms (4 bodies of work/6 papers). Several studies targeted specific surgical specialties, for example orthopedics and cardiothoracics [61] or pediatric surgery [56,60]; and others sought to include a diverse range of specialties e.g. Ierano, Thursky (59). Seven papers, from six bodies of work, focused purely on surgical prophylaxis. The remainder focused on a range of aspects relating to surgical antimicrobial prescribing during the surgical pathway, for example, the impact of culture and team dynamics on the ward round; and features that characterize antimicrobial decision-making in the Surgical Intensive Care Unit (SICU)–see Table 3 for a full breakdown.

## Critical appraisal and analysis

For the purpose of this meta-ethnography, nine of the 14 included papers were deemed key papers [26–28,57–61,64], meaning we perceived them to be conceptually rich with the potential to make an important contribution to the synthesis. The remainder were classified as satisfactory [56] as they were perceived to be less valuable than key papers–e.g. less conceptually rich but likely still relevant; or questionable [62,63,65,66] as their likely contribution, to answering the research question in this meta-ethnography, was uncertain.

Forty-eight concepts were identified from the 14 included studies, which provided the eight overarching concepts (see Table 4). These overarching concepts interacted to varying degrees and, interestingly, there was not consensus among stakeholders regarding an order of importance. Patient representatives were anxious that 'discontinuity of care: physical and team structures create silos and barriers to communication and workflow' represented a serious risk to patient safety; whereas several healthcare workers felt that 'hierarchy' was a major determinant of APB and as such would be the key to unlocking many opportunities for change.

The overarching concepts are explained in detail below. However, it is only by understanding how the concepts interact that we will be able to advance our thinking and find workable solutions [48]. An elaboration of the likely inter-dynamics between overarching concepts is presented in the discussion section, where we move beyond the primary data.

## Hierarchy

In surgery, autonomous decision-making by individual actors is clear in the influence of consultant surgeons on the team [26]. On this basis, the prescribing behavior is defined and

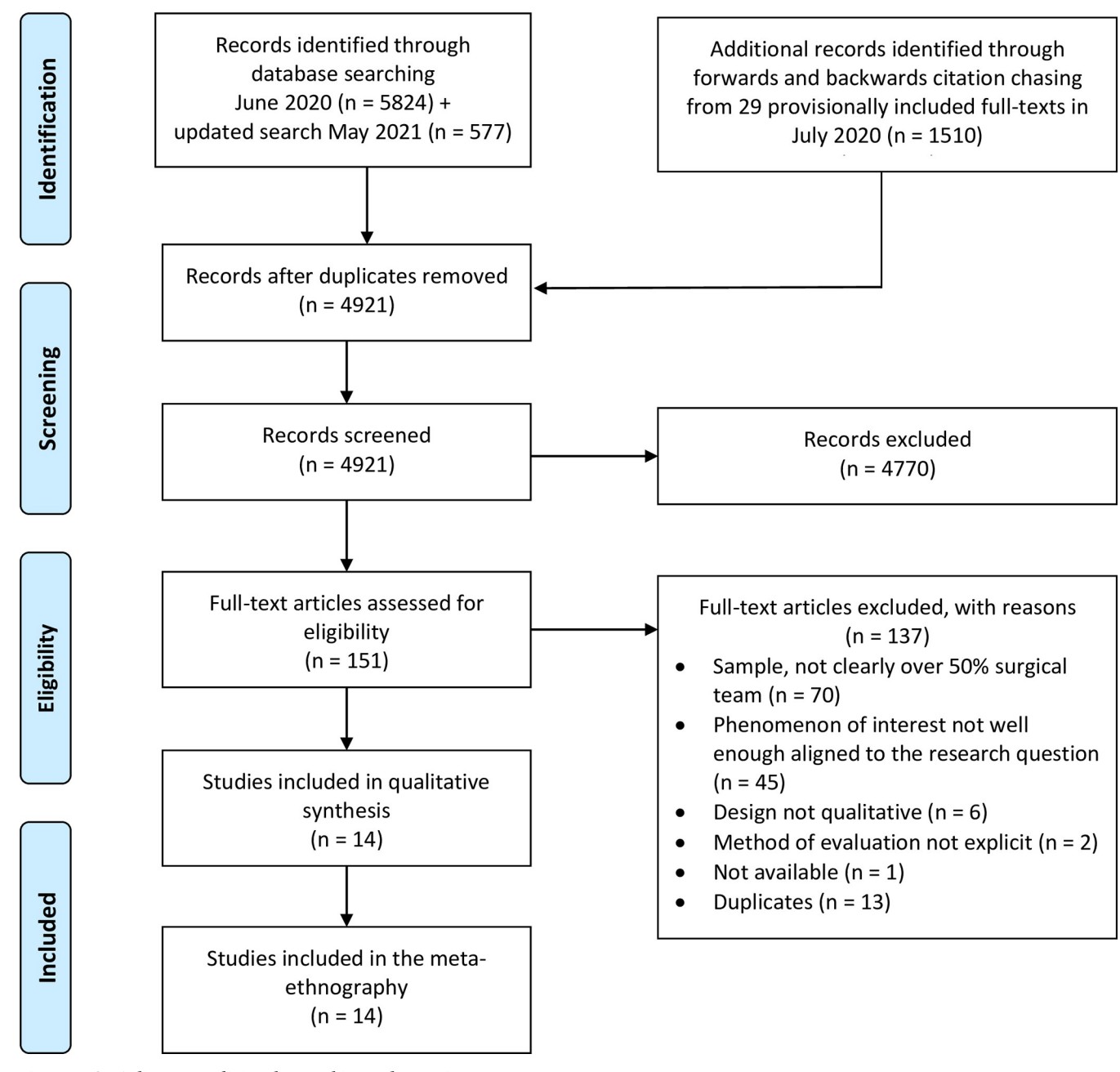

**Fig 1. PRISMA diagram outlining the searching and screening process.**

regulated at a senior level, dependent on the surgical consultant's preference, as opposed to guideline recommendations [59]. The vertical structure of the surgical team (surgical doctors with different levels of training, experience and responsibility i.e. the consultant, specialist registrar, and more junior trainee doctors) leaves little room or opportunity for input from other team members, particularly during ward rounds [26]. The surgeons, though not fully engaged with the antibiotic decisions made for their patients, remain the leaders in their specialty and engaging with them in antimicrobial stewardship will be tantamount to engaging with their

**Table 3. Study characteristics of the papers to be synthesized.**

| | Source paper (n = 14) | Study type | Institutional setting | Country | Data collection and sample | Surgical specialty | Aim |
|---|---|---|---|---|---|---|---|
| 1 | Giusti et al. (2016) [56] | Mixed-methods (focus groups and a self-administered questionnaire) | Three tertiary care children's hospitals with 192 beds; 607 beds; and 442 beds | Italy | Focus groups with 33 healthcare professionals including 15 surgeons, 10 nurse coordinators, and 8 anesthesiologists; and a self-administered questionnaire completed by 357 surgical healthcare professionals (82% response rate) | Pediatric surgery | To investigate barriers and tot describe the attitudes of healthcare professionals regarding SAP in three Italian children's hospitals |
| 2 | Charani et al. (2017) [27] | Qualitative | One multi-site teaching hospital with 1300 beds | England (London) | Ethnographic observation: 30 acute and elective ward rounds (over 100 hours) involving six surgeons and their teams; and observation on the ward and in handover and team meetings (over 50 hours). Semi-structured interviews: 13 key informants were interviewed, including 5 consultant surgeons, 3 registrars, 2 nurses, 2 junior doctors and 1 ward pharmacist | Adult emergency and elective surgery (including patients under the care of the surgical team for non-surgical care, for example cholangitis and cholecystitis) | To investigate the impact of culture and team dynamics of the surgical ward round on antibiotic decision making |
| 3 | Broom et al. (2018) [28] | Qualitative | One tertiary referral, teaching hospital with 450 beds | Australia (New South Wales) | Semi structured interviews with 20 doctors (17 surgeons and 3 anesthetists). Of the 17 surgeons, 10 were senior and 7 were junior. The 3 anesthetists included 2 senior and 1 junior participant | General surgery, neurosurgery, orthopedic surgery, colorectal surgery, urology, transplant surgery, cardiothoracic surgery, vascular surgery and renal surgery | To explore through semi-structured interviews the experiences and perceptions of surgeons and anesthetists around SAP prescription and administration, to provide insight into social factors which may be barriers to implementation of evidence-based practice in this area of antibiotic use |
| 4 | Broom et al. (2018a) [57] | Qualitative | One tertiary referral, teaching hospital with 450 beds | Australia (New South Wales) | Semi structured interviews with 20 doctors (17 surgeons and 3 anesthetists). Of the 17 surgeons, 10 were senior and 7 were junior. The 3 anesthetists included 2 senior and 1 junior participant | General surgery, neurosurgery, orthopedic surgery, colorectal surgery, urology, transplant surgery, cardiothoracic surgery, vascular surgery and renal surgery | To examine in depth the perspectives of surgeons and anesthetists on interpersonal and cultural SAP prescribing influences |
| 5 | Charani et al. (2019) [19] | Qualitative | One multi-site teaching hospital with 1300 beds | England (London) | Ethnographic observation (total = 500hrs) including: 30 surgical ward rounds; 22 medical ward rounds; and observation of routine healthcare worker practices on the wards. Face-to-face interviews with 23 key informants including surgeons, medical consultants, trainee doctors, nurses, and pharmacists (14 from surgery; and 9 from medicine) | Acute surgery (wards with a high percentage of elective and non-elective admissions) | To investigate and compare cultural determinants of antibiotic decision-making in acute medical and surgical specialties |
| 6 | Broom et al. (2019) [58] | Mixed-methods (pre- and post-intervention audit; and qualitative assessment) | One hospital with 450 beds | Australia (Queensland) | Quantitative: 23 patients before and 22 patients after the intervention were included. Qualitative: semi-structured interviews with 18 healthcare professionals (4 nurses; 2 pharmacists; and 12 doctors—5 senior and 7 junior) | General Surgery (specifically focusing on complicated intra-abdominal infections requiring definitive surgical source control) | To compare antibiotic prescribing patterns before and after a multifaceted persuasive intervention addressing social factors likely to impact antibiotic duration in patients with source-controlled intra-abdominal infections; and to conduct a qualitative assessment to identify which factors enhanced or detracted from the perceived success of the intervention (i.e. to identify which aspects of the intervention are likely to have been effective) |

*(Continued)*

**Table 3.** (Continued)

| | Source paper (n = 14) | Study type | Institutional setting | Country | Data collection and sample | Surgical specialty | Aim |
|---|---|---|---|---|---|---|---|
| 7 | Ierano et al. (2019) [59] | Qualitative | Three tertiary public and private hospitals | Australia | Fourteen focus groups and one paired interview with 77 surgical healthcare workers (6 surgical registrars and residents; 13 theatre nurses; 10 anesthetists; 40 surgeons; and 8 pharmacists) | Orthopedics; general surgery; cardiac surgery; vascular surgery; and plastic and reconstructive surgery | To identify barriers and enablers of appropriate SAP prescribing and evidence-based guideline compliance; and to compare the perceptions of health professionals in surgical specialties across both public and private hospital settings regarding these barriers and enablers |
| 8 | Malone et al. (2020) [60] | Qualitative | One quaternary-care children's hospital | United States of America | Five semi-structured focus groups with 23 surgeons | Pediatric surgical specialties including: interventional cardiology; otolaryngology; orthopedic surgery; cardiothoracic surgery; and general surgery | To understand the factors that contribute to pediatric surgeons' decisions regarding the use of perioperative antibiotic prophylaxis |
| 9 | Peel et al. (2020) [61] | Qualitative | One public, adult only, quaternary, university-affiliated hospital | Australia (Melbourne) | Focused ethnographic observation (20 hours in the preadmission clinic; 25 hours in the operating room; and 13 hours on postoperative ward rounds); and face-to-face semi-structured interviews with 6 senior clinicians (2 surgeons and 4 anesthetists) | Orthopedic surgery; and cardiothoracic surgery | To describe the phenomenon of and culture of antimicrobial decision making in two surgical specialty units (orthopedic and cardiothoracic surgery) |
| 10 | Rynkiewich et al. (2020) [62] | Qualitative | Two teaching hospitals: a private academic medical center (24 bedded 'open' SICU); and a public teaching hospital (14 bedded 'open' SICU) | United States of America (Mid-Western) | Ethnographic observation on 40 ward rounds (over 160 hours); and 10 semi-structured interviews with 10 of the ward round participants (4 SICU attending surgeons; 2 SICU attending anesthesiologists; 1 SICU attending pulmonologist; 2 surgery fellows; and 1 pulmonology fellow) | Surgical intensive care unit (SICU). *At both hospitals included in this study, the primary decision maker for the patient was acknowledged to be the surgeon that had operated on the patient and retained primary care responsibilities* | To explore the features which characterize antibiotic decision making in the SICU |
| 11 | Singh et al. (2021) [63] | Qualitative | Two university hospitals (South Africa: a 950-bed government-funded tertiary hospital which also provides non-tertiary services to the local population; and India: a 1350 bed not-for-profit charitable tertiary center) | South Africa and India | Ethnographic observation of clinical practices (210 hours); 6 patient case studies; and face-to-face interviews with 105 healthcare professionals and 14 patients | Adult specialties including: cardiovascular surgery; thoracic surgery; and gastrointestinal surgery | To investigate the drivers for infection management and antimicrobial stewardship (AMS) across high-infection-risk surgical pathways |
| 12 | Bonaconsa et al. (2021) [64] | Qualitative | One 950-bedded tertiary public and government-funded referral university hospital | South Africa (Cape Town) | Ethnographic observation of clinical practices (190 hours: 138 hours in India; and 72 hours in South Africa), interviews with HCPs (44 India, 61 South Africa), patients (6 India; 8 South Africa) and case studies (4 India; 2 South Africa) | Cardiothoracic, gastrointestinal acute care, and gastrointestinal colorectal surgical units | To study how surgical team dynamics and communication patterns influence infection-related decision making using innovative visual mapping alongside traditional qualitative methods |

**Table 3.** (Continued)

| | Source paper (n = 14) | Study type | Institutional setting | Country | Data collection and sample | Surgical specialty | Aim |
|---|---|---|---|---|---|---|---|
| 13 | Broom et al. (2021) [65] | Mixed-methods (quality improvement intervention with qualitative assessment) | Three hospitals (one regional; and two metropolitan) | Australia | Quantitative: SAP prescribing decisions for 1757 patients undergoing general surgical procedures from three health services were included. Six bimonthly time points, pre-implementation and post implementation of the intervention, were measured. Qualitative: individual semi-structured interviews with 29 clinical team members from across the three sites—25 doctors (10 senior surgeons, 1 senior Infectious diseases doctor; and 14 junior doctors with varying levels of experience) and 4 pharmacists | General surgery | To assess an intervention for surgical antibiotic prophylaxis improvement within surgical teams focused on addressing barriers and fostering enablers and ownership of guideline compliance |
| 14 | Khan et al. (2021) [66] | Mixed-methods (self-administered questionnaire and focus groups) | One large teaching and tertiary referral hospital | India (Western Uttar Pradesh) | Quantitative: pre-test questionnaire with 6 closed questions regarding SAP Quantitative: 28 focus groups and 16 paired interviews with: general surgeons (n = 39; 21%), gynecologists (n = 33; 17.9%), orthopedic surgeons (n = 43; 23.3%), pediatric surgeons (n = 2; 1%), plastic surgeons (n = 6; 3.2%), neurosurgeons (n = 4; 2.1%), otorhinolaryngology (n = 9; 4.9%), and anesthesiologists (n = 48; 26%). Most of the participants were junior residents (136; 73.9%) | Orthopedics, general surgery, obstetrics and gynecology, otorhinolaryngology, plastic surgery, pediatric surgery, and anesthesiology | To assess the knowledge and compliance rate for SAP guidelines among various surgical specialties and those involved in providing SAP |

SAP: Surgical antimicrobial prophylaxis; SICU: Surgical intensive care unit.

entire team [26]. They have the power to influence the behaviors of their entire team (as the junior team must ratify any decisions verbally or via messages with the senior team) [26].

SAP decisions are embedded within hierarchical relationships and decisions are influenced by both intra (i.e. within the surgical team) and inter (e.g. between surgeon and anesthetist) specialty hierarchies [57,66]. This may result in interpersonal risk and disempowerment of individuals positioned in situations necessitating challenge to established hierarchies to bring about change [57]. Junior doctors consider adhering to decisions made by their seniors as mandatory; challenging decisions made by more senior doctors is perceived as conferring interpersonal risk [57], potentially jeopardizing their career and/or relationship with the surgeon, and is therefore not considered to be worth it [59,66]. Furthermore, surgeons may not speak up against their colleagues or seniors regarding antibiotic management due to fear of negative consequences [59].

Adding to the complexity, surgeons, regarded as being at the top of the hierarchy, may be reluctant to receive advice and feedback from those not at a similar level of seniority; and feedback is likely to have a higher impact if delivered by health professionals of 'similar status', i.e., anesthetic and infectious diseases consultants, as opposed to pharmacists and nurses [59]. Antimicrobial stewardship advice from a lower hierarchical level may be attributed less value than advice from higher levels of seniority within the antimicrobial

**Table 4. Translation of second order concepts, from the 14 included papers, into overarching (third order) concepts.**

| Overarching concept* | Second order concepts contributing to the overarching (third order) concept | Papers that include the second order concept (references in bold are key papers) |
|---|---|---|
| Hierarchy | • Hierarchical relationships and power <br> • Uptake in knowledge and change in practice <br> • Senior ownership and engagement <br> • Individualism <br> • Hierarchy <br> • SAP decisions are entrenched within hierarchical relationships <br> • Responsibility and power dynamics | **Broom et al., 2018a [57];** <br> **Broom et al., 2019 [58];** <br> Broom et al., 2021 [65]; <br> **Charani et al., 2019 [19];** <br> **Ierano et al., 2019 [59];** <br> Khan et al., 2021 [66]; <br> **Peel et al., 2020 [61]** |
| Fear drives action | • Fear of infectious complications drives overuse of SAP <br> • Ownership of surgical risk <br> • Fear and the need and expectation to intervene <br> • Fear and the impetus to "do something" <br> • Fear is a driving factor for the prolongation of SAP <br> • Surgeons fear adverse patient outcomes <br> • Antibiotics a safety net | **Broom et al., 2018 [28];** <br> **Broom et al., 2018a [57];** <br> **Charani et al., 2017 [27];** <br> **Charani et al., 2019 [19];** <br> **Ierano et al., 2019 [59];** <br> Khan et al., 2021 [66]; <br> Singh et al., 2021 [63] |
| Deprioritized | • SAP decision making is a peripheral issue <br> • Other tasks are prioritized above antibiotic decision making <br> • Antibiotic management is a peripheral issue <br> • Antimicrobial stewardship has a low priority <br> • SAP is a low priority for surgeons <br> • SAP prescription is not considered a priority <br> • Antibiotic prescribing is not prioritized and is rarely discussed | **Broom et al, 2018 [28];** <br> **Broom et al., 2018a [57];** <br> **Charani et al., 2017 [27];** <br> **Charani et al., 2019 [19];** <br> **Ierano et al., 2019 [59];** <br> Khan et al., 2021 [66]; <br> **Peel et al., 2020 [61]** |
| Convention trumps evidence: skepticism and improvisation limit the impact of surgical antibiotic prophylaxis (SAP) evidence-based guidelines, and social norms shape action | • Improvisation behaviors <br> • Trust, disagreement and clinical judgement <br> • Guideline limitations and autonomy <br> • Gaps warrant exceptions <br> • Skepticism | **Broom et al., 2018 [28];** <br> Giusti et al., 2016 [56]; <br> **Ierano et al., 2019 [59];** <br> Khan et al., 2021 [66]; <br> **Malone et al., 2020 [60]** |
| Complex judgements | • Tolerance of uncertainty <br> • Lack of feedback <br> • Antibiotic decision making <br> • Risk assessment <br> • Overvalued perception of the benefit of antimicrobials <br> • Antibiotics a conservative intervention | **Charani et al., 2019 [19];** <br> Giusti et al., 2016 [56]; <br> **Ierano et al., 2019 [59];** <br> **Malone et al., 2020 [60];** <br> **Peel et al., 2020 [61];** <br> Rynkiewich et al., 2020 [62] |
| Discontinuity of care: physical and team structures create silos and barriers to communication and workflow | • Separation of the infectious diseases team <br> • Constant state of flux <br> • Time <br> • Not part of the team <br> • Physical work environment <br> • Physical proximity <br> • Consulting service | **Broom et al., 2018a [57];** <br> **Charani et al., 2017 [27]; Charani et al., 2019 [19];** <br> **Ierano et al., 2019 [59];** <br> **Peel et al., 2020 [61];** <br> Rynkiewich et al., 2020 [62]; <br> Singh et al., 2021 [63] |
| Team dynamics and interactions create unrealized potential | • Unrealized potential <br> • Relationship dynamics <br> • Inter-specialty team dynamics <br> • Collaboration and challenge between diverse practice groups | **Bonaconsa et al., 2021 [64];** <br> **Broom et al., 2018a [57];** <br> **Malone et al., 2020 [60];** <br> Rynkiewich et al., 2020 [62] |

*(Continued)*

**Table 4.** (Continued)

| Overarching concept* | Second order concepts contributing to the overarching (third order) concept | Papers that include the second order concept (references in bold are key papers) |
|---|---|---|
| Practice environment: organizational features and resources nudge decision-making | • Absence of structured handover tools<br>• Organizational and structural determinants can promote overuse of SAP<br>• Unavailability or interrupted supply of antimicrobial agents<br>• Structural issues<br>• Private context | **Bonaconsa et al., 2021 [64];**<br>Giusti et al., 2016 [56];<br>Khan et al., 2021 [66];<br>**Malone et al., 2020 [60];**<br>Singh et al., 2021 [63] |

*See the results section for a detailed explanation of the overarching concepts. SAP: Surgical antimicrobial prophylaxis.

stewardship team; therefore, the power of antimicrobial stewardship advice of discordant seniority to alter decision making by a surgeon or anesthetist is potentially limited [57].

Targeting stewardship interventions solely at junior surgeons is unlikely to be helpful, as their prescribing is heavily regulated by their seniors' preferences, as opposed to externally received advice advocating guideline compliance [59]. Uptake of knowledge and change in practice is also often determined by hierarchical influences within a specialty service [58]. To generate changes in prescribing practices, acceptance of interventions from the most senior level is thus required [59]. This is because the ownership of the change process by the surgical service, and leadership by senior surgical members, increases the profile and efficacy of interventions [58]. Moreover, senior ownership of and engagement with an intervention are indicators for its likely success [65].

## Fear drives action

What is considered unique in surgery is that a patient has to be well enough to be able to undergo an operation, therefore any deterioration postoperatively is assumed to be a consequence of the surgery, and the decisions of the surgeon, and not the patient's underlying illness i.e. the overwhelming responsibility for the patient remains with the surgeon [27]. The impetus to "do something" is greater in surgery (than medicine), as "patients are not allowed to die" [26]. These concerns drive a more conservative approach to antibiotic decision-making leading to unnecessary and prolonged courses of antibiotics [27]. The need and expectation to intervene means that often antibiotics are initiated for patients with little or no evidence of infection, but a high plausibility of infection in the minds of the surgeons [27]. This process is rationalized by the surgeons as being an extension of their roles as 'interventionists' [27]. In the absence of concrete evidence of infection, what drives antibiotic prescribing and decision-making is fear, of the risk of possible infections, and thus worse patient outcomes; and the risk of blame [27]. This practice drives inappropriate antibiotic use, when it is not indicated by guidelines, and the addition of antibiotic doses over and above usual guideline recommendations [28], particularly in the postoperative phase [27]. A focus on starting, but not on reviewing or stopping, treatment can lead to unnecessary and prolonged courses of antibiotics [26]. Additionally, a surgeon's negative experiences such as previous complications [57] and litigation (or fear of litigation) appear to encourage the over-prescribing of surgical antimicrobial prophylaxis (SAP) to avoid a 'backlash' for post-operative complications and surgical site infections from many sources: the hospital, surgical colleagues, other medical professionals and health insurance companies [59].

## Deprioritized

The primary responsibility for the patient remains with the surgeons [rather than other members of the multi-disciplinary team, for example anesthetists or nurses] [27]. However, the surgeons identify their main role (object of the work activity) to be addressing the patient's surgical problems [26,27]. All other tasks during the care process, including antibiotic management, are thus peripheral to this and may be missed by them [27]; and antibiotic prescribing is rarely discussed between senior and junior surgical team members, or with patients [61].

On the ward round, the lack of priority given to antibiotic decision making is compounded by a lack of expertise [27], resulting in the responsibility for antibiotic decisions being commonly delegated to junior members of the surgical team, who are still learning the prescribing activity, leading to more "defensive" approaches to prescribing [61]. In other words, more antibiotics are prescribed, and for longer periods, to protect the caregiver from criticism or complaint (see fear drives action).

In the operating theatre, prescribing is, however, senior led [61]. Yet,in this context it also remains a peripheral issue due to the complexity of care processes occurring and decision making can break down, especially when team members (surgeon/anesthetist) are not known to each other; and in emergency situations where both surgeons and anesthetists are occupied with other tasks which are prioritized [28,57].

## Convention trumps evidence: Skepticism and improvisation limit the impact of surgical antibiotic prophylaxis (SAP) evidence-based guidelines, and social norms shape action

In the context of surgical antimicrobial prophylaxis, social norms shape action and result in behavior that does not conform to evidence-based practice [28]. While surgeons are aware of guidelines, the need to comply with them does not perceivably drive or regulate current practices [59]. Inconsistent guideline implementation is seen as an accepted part of current practice–where prescriber autonomy overrules guideline compliance and social codes of prescribing reinforce established practices [59].

Disagreement of surgical staff with surgical antimicrobial prophylaxis guidelines is a barrier to adherence [56]. In addition, surgeons may be distrustful of the literature and guidelines [60]. They perceive the existence of many gaps in the current evidence [59]; and consider guidelines to be too general and inadequate to account for the broad range of surgical procedures and any environmental and patient-specific factors [66]. Furthermore, some surgeons question the applicability of guidelines to their patient population, commonly mentioning the uniqueness of each patient and therefore an unwillingness to change their antibiotic practices without first having evidence of high quality and direct relevance to their specific subspecialty and/or a specific procedure [60]. Among surgeons, prescriber autonomy and discretion are considered to be paramount in SAP decision making [59]; and improvisation behaviors are driven by: concern around adverse patient outcomes; a sense of benevolence towards the patient (held by the surgeon)–conferring a sense of having done everything possible to prevent an infectious complication; an internalized sense of what is perceived conventional practice for a particular operation; and the perceived extra layer of safety for the surgeon (safety from both litigation and also from personal responsibility for a complication) [28]. There is a recognized discord between the evidence and these improvisation practices but, despite this, they confer a sense of reassurance to the surgeon [28].

Surgical staff are more inclined to trust guidelines if they are developed by a multidisciplinary group of peers with recognized scientific and methodological knowledge and experience of the local context [56].

## Complex judgements

Surgeons evaluate the risks and benefits of antibiotic use and weigh many factors, for example patient risk of infections and their previous learning and experience [60]. However, an over-valued perception of the benefit of antimicrobials drives overuse [61]; and knowledge of hospital data on quality of SAP administration, and outcomes for surgical procedures (such as the incidence of surgical site infections) is often limited [56]. Antibiotic decision-making is focused more on prevention than on treatment of infections [26]. Although surgeons understand the long-term outcome of antibiotic resistance, from unnecessary antibiotic use, the negative consequences of a surgical site infection and/or reputational damage are much more concerning and prominent to them [60]. Surgeons believe that overprescribing antibiotics is a mechanism to reduce the risk of SSI for the patient and to improve hospital performance measures but they discount the potential harm of the antibiotics (i.e. antibiotic resistant bacteria, *Clostridium difficile* infection) [60]; and perceive their contribution to AMR to be minimal compared to other contexts, such as critical care units, primary care and nursing homes [59,60]. In sum, the level of autonomy (dominance of the individual approach) among surgeons, together with fear of negative outcomes in surgery, leads to less tolerance of uncertainty [26].

## Discontinuity of care: Physical and team structures create silos and barriers to communication and workflow

Communication pathways are influenced by the physical work environment [61]. The surgical team is constantly split between theatres, clinics and the ward [27]. Hence, there is limited time for in person communication, and a lack of clarity about the responsibility for antibiotic management of patients, and antibiotic prescribing takes place in the context of disjointed information [27](adding complexity to antibiotic decision making–see complex judgements). This leads to poor continuity of care and sub-optimal antibiotic management [27]. Due to their other obligations, senior surgeons are often absent from the ward, leaving junior staff to make complex medical decisions which results in defensive antibiotic decision-making (see fear drives action), leading to prolonged and inappropriate antibiotic use [27]. There is little opportunity for forward planning and a heavy reliance on digital rather than in-person communication [26]. The different professions do get involved in the surgical patient pathway, but they tend to work in silos (frequently within the physical divisions), with few communication opportunities between them [26] (see team dynamics create unrealized potential). Thus, multi-disciplinary teamwork is not easily practiced in such a context [26]. Antimicrobial stewardship teams and infectious diseases specialties are not physically [57] or figuratively [63] considered to be part of the surgical team and as such their recommendations, for surgical patients, are to be considered but not mandatorily enforced [59]. The final decision remains with the surgeon and the surgical teams usually recognizes the input. However, it is viewed as a consulting service, and the antibiotic decision-making remains within the surgical team [59,63]. The lack of integration of the stewardship or infectious diseases teams into the surgical environment is a barrier to antimicrobial stewardship programs [61].

## Team dynamics and interactions create unrealized potential

Collaboration and challenge between diverse practice groups may influence antibiotic decision making and clinician autonomy [62]. In theatres, the relationship dynamics between the surgeon and the anesthetist determine the appropriateness of SAP, particularly operative risk ownership but also familiarity and cohesiveness [57]. Additionally, surgeon-stewardship team

dynamics impact the way antibiotics are prescribed–surgeons feel solely responsible for adverse outcomes in their patients and think it's an issue their stewardship colleagues do not have to address–poor relationships can interfere with the antibiotic stewardship team's ability to make recommendations about the most appropriate practices [60].

The surgical ward round, though attended by different professionals, remains a medium of communication between registrars and consultants, with little interaction with the patient or other healthcare professionals [64]. Where members of the team position themselves on the round is a predictor of their participation [64]. Although not everyone is expected, nor allowed, to contribute equally, discussions predominantly engage members who are physically and figuratively in the middle (generally consultants and registrars) [64]. Remaining mostly on the outer boundaries, nurses face communication limitations and are not always fully engaged in decision-making [64]. This is despite the critical information that they could provide to patient care [64]. Unrealized potential exists for nurses to have a more active role in antimicrobial stewardship and infection prevention and management to prompt antibiotic review, especially intravenous to oral switch, as well as to monitor for adverse drug effects [64]. The leadership style of the surgeon-consultant leading the round impacts on ward round dynamics (see hierarchy); and team-focused consultants can facilitate intentional and active engagement with the patients and the wider ward round team, to harness the unrealized potential [64].

## Practice environment: Organizational features and resources nudge decision-making

Structural issues at the hospital and/or departmental level can encourage the overuse of SAP in surgical patients [60] either in terms of administering antibiotics in procedures where SAP is not indicated; choosing a second choice (broader-spectrum) antibiotic; or administering antibiotics for more than 24 hours [56]. For example, structured workflow e.g. out-of-date order sets [60]; overcrowding of patients' rooms; proximity to other potentially infectious patients; absence of clean patient routes, such as elevators restricted for the operating room; and lack of hand hygiene facilities in every room [56]. The absence of tools can also impact care. For example, the absence of structured handover tools can mean that care delivery is influenced by factors such as high patient volumes; rushed ward rounds; various handover and leadership styles by registrars and consultants; other surgical priorities; and the rotation of registrars through specialties [64]. Additionally, the unavailability or interrupted supply of certain antimicrobial agents is a major logistical problem that puts the surgeon in a position to prescribe antimicrobial agents from the limited options available [66]. In the private context, treatment decisions (including antimicrobial therapy) may end up being tailored to the patient's financial capability; or being more conservative as clinicians are disempowered to negotiate with patients that can choose to go elsewhere for care if they are not satisfied [63].

## Line-of-argument synthesis

From the translation we were able to develop a line-of-argument synthesis [47], which posits that social and structural mediators influence individual complex antimicrobial judgements and currently skew practice towards increased and unnecessary antimicrobial use which results in increased patient harm, AMR and costs. This is represented by our conceptual model (Fig 2).

The model illustrates the complex nature of surgical antimicrobial prescribing. The two left hand boxes show how external social and structural mediators influence complex judgements made by an individual prescriber. In the central box, the individual's behavior and beliefs are

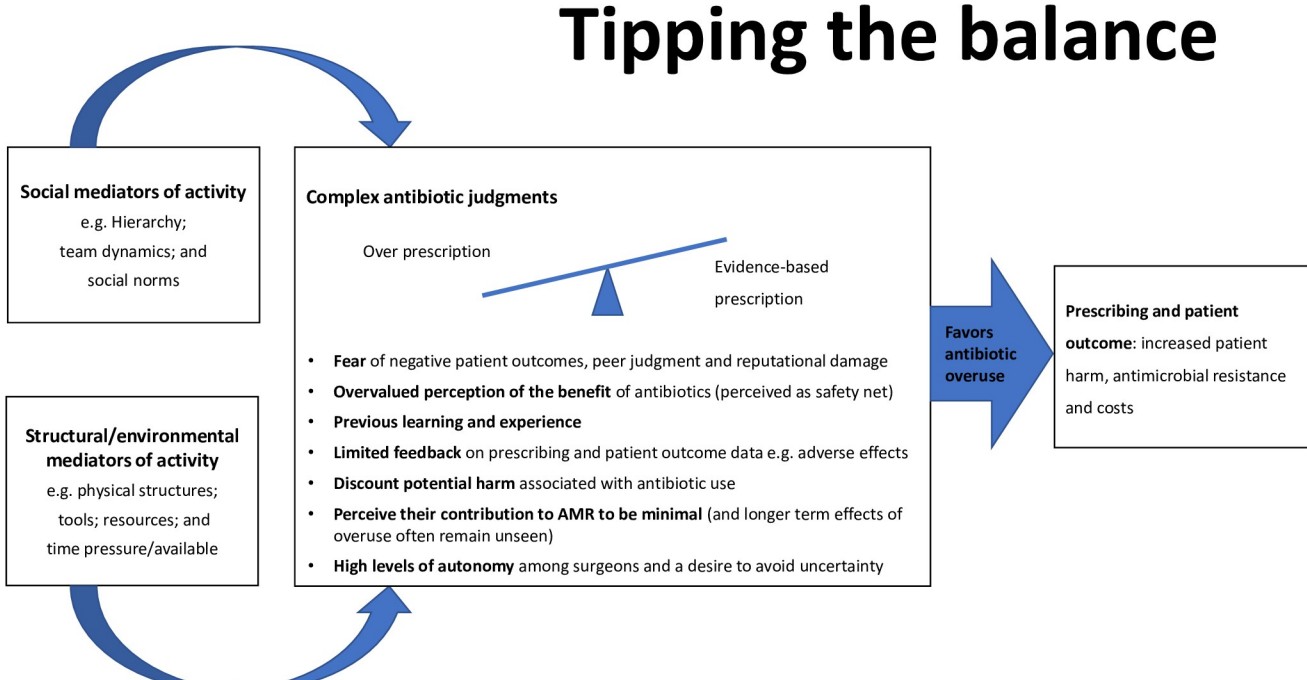

**Fig 2. Conceptual model showing how social and structural mediators influence individual complex judgements about whether to prescribe antimicrobials for surgical patients, currently tipping the balance towards unnecessary antimicrobial use and resulting in increased patient harm, AMR and cost.**

shaped and perpetuated by organizational factors, which are currently weighted towards over prescribing, as exemplified by the fulcrum. The challenge is for healthcare workers to tip the balance such that evidence-based prescribing becomes the default and this model provides insights on how this may be achieved (see implications for policy and practice).

Stakeholder engagement favored this model (that was iteratively developed with their input) for a variety of reasons including: its simplicity and clarity; its ability to capture the complexity involved (thus demonstrating a level of empathy for practitioners); and its focus on the impact of the organization (moving ownership to the organization rather than focusing on individuals). Importantly, all surgical team members of the stakeholder group, who are the target audience for change initiatives, were amenable to and understood both our line-of-argument and conceptual model.

## Discussion

This study aimed to synthesize qualitative studies that explore surgical antimicrobial prescribing behavior (APB) to explain how and why contextual factors act and interact to influence APB amongst surgical teams, and to develop new clinically applicable theory to advance understanding. Our analysis makes several original contributions. Firstly, as far as we know, it is the first qualitative synthesis addressing this question. Secondly, and more importantly, this work highlights the major influence of both social and structural factors on what are very complex individual prescribing decisions. This is fundamentally important as, in order to change the status quo, research needs to render visible the effects of contextual factors on APB and on the more distal health outcomes.

The eight overarching concepts identified via synthetic processes resonated with stakeholders–including practicing clinicians and patients–and, although they exhibit similarities with

studies that explore hospital prescribing behavior more generally [6,8,67,68], they also highlight the surgical context as different which supports our, and others', stance that a nuanced approach to improvement will be key [69,70].

Among surgical specialties, there is typically a very steep hierarchy [26,71]. This can limit input from the wider healthcare team, such as from those at the bedside providing direct patient care and from infection experts, which represents a missed opportunity [62,64]. Furthermore, uptake of knowledge and change in practice is often determined by the hierarchy as exemplified by a quote from a doctor involved in a surgical quality improvement intervention: '*When your boss says, "This is what I want done," that's how you start to go, "Right, well this is not just something that the fairies are telling me must happen, but this is something that my boss wants," and. . . that tends to sometimes just get it to happen*' [58]. The comment epitomizes the unquestioning fashion with which juniors follow the the consultant surgeon (who has the clinical decision making power to make things happen), and shows how other players (fairies) may be ineffective to drive change. In this way, the hierarchy can perpetuate rituals (such as non-evidence based, prolonged surgical prophylaxis); reinforce mistrust in the current evidence-base; and maintain a status quo in which antimicrobials are not prioritized, underscoring the importance of the surgical milieu vis-a-vis antimicrobial misuse [6,72].

The physical dispersion of the surgical team (between theatres, clinics and the surgical wards) adds another layer of complexity, creating communication challenges which threaten the continuity of patient care [27,61]. Time for ward rounds is usually limited as a consequence of the surgical teams' competing obligations [26,73]; and tools to structure the process are often absent [64], creating an environment that does not support complex antimicrobial decision-making. Junior members of the surgical team, who are still learning the prescribing activity, may be left to make complex antimicrobial prescribing decisions [27]. This understandably leads to a more defensive approach [74] which manifests as more and/or longer, potentially unnecessary, antimicrobial [treatment] courses [72]. Additionally, patients are almost always omitted from the decision-making process [4,61,64] echoing a recent scoping review that highlights the need for more patient (and carer) involvement in antimicrobial stewardship across the surgical pathway [75].

Within the confines of the time pressured, fast flowing environment, it is clear that surgeons weigh many factors when prescribing antimicrobials; they are not naïve to the threat of AMR [60]. However, for the surgical team, the risk of complications and the resultant personal blame are a far more concrete threat than that of possible AMR; and there is a perception that patients [particularly elective patients] are not allowed to die [26]. Antimicrobials are viewed as a safety net and means to reduce uncertainty and risk [26,60,72]. This stance is perpetuated by the lack of feedback on prescribing decisions and patient outcomes (such as drug related side-effects), including the fact that AMR usually plays out downstream hiding lines of accountability [as discussed by Livorsi, Comer (8)], potentially creating a feedback sanction (i.e. a cognitive error whereby a significant time delay or complete lack of feedback on the consequences of a decision reinforces the behavior).

## Implications for policy, practice and research

Our conceptual model (Fig 2) can be used to provide insight into how we might 'tip the balance' towards more judicious and evidence-based antimicrobial use across surgical specialties. Understanding the dominant influence of the hierarchy on surgical APB will enable those designing quality improvement interventions to embed components that address this mediator. For example, we would encourage stewardship teams to engage directly with surgeons, at multi-disciplinary meetings and educational forums, and then to communicate consensus decisions to less powerful junior staff. Ownership of the change process, and leadership, by

senior surgical staff increases the profile and efficacy of interventions [58] indicating that co-creation will be instrumental in fostering change (see theoretical implications).

The important role of team dynamics suggests that stewardship teams should prioritize building relationships with consultant surgeons and chiefs of surgery. They must also be mindful that much of the 'errant' prescribing comes from a good place i.e. a sense of benevolence towards the patient [28]. Rather than criticizing surgical teams for poor performance, stewardship teams must step up as infection experts, to present convincing arguments, backed by high-quality evidence from relevant patient populations, to demonstrate benefits and lack of unintended consequences for patients, thereby challenging entrenched behaviors. Additionally, feedback on clinical outcomes (e.g. SSIs and adverse drug reactions), and prescribing, should be provided to individual surgeons, benchmarked with their peers, to demonstrate (and reassure) that over-prescribing does not deliver better outcomes.

Greater integration of the multidisciplinary team, including infection experts and those providing bedside care (with up-to-date knowledge of the patient's clinical status), is also likely to improve antimicrobial stewardship in surgery. However, we recognize the dichotomy between asking specialty trainees e.g. infection registrars, nurses and pharmacists to do more (to support antimicrobial stewardship) and the steep hierarchy–whereby surgeons may be reluctant to receive advice and/or feedback from health professionals of a different and/or lower status [59], thus lessening the impact. Furthermore, healthcare professionals cannot always just take on more. To be effective, nurses and/or pharmacists may need to be assigned a designated stewardship role that provides them with the necessary time and empowerment to intervene, without fear of repercussions. We would also advocate for greater inclusion of patients and carers in stewardship dialogue and anticipate that this would favor more rational antimicrobial use–helping to 'tip the balance'.

Some of the structural mediators of APB may be best addressed through the implementation of new systems. For example, ward round checklists have been shown to improve communication and documentation [76]. Additionally, a review of working patterns/job plans may support improved communication and workflow. For example, consultant surgeons need protected time to conduct thorough ward rounds–for the benefit of patients, and to positively impact on workplace culture. Likewise, the integration of an infection expert at designated, mutually convenient, times would provide valuable support for non-infection experts when making complex antimicrobial judgements.

Although the meta-ethnography provides insights, it also raises questions and identifies gaps in what is understood. Surgeons are in charge; however, they rarely prioritize leadership vis-à-vis antimicrobial management. This juxtaposition is important, but how to tackle it is unclear and warrants further research. We need to better understand the context for change. What would make the surgical team prioritize antimicrobials? Why are surgeons distrustful of literature and guidelines? What is the most effective way to integrate wider members of the multi-disciplinary team? Would it be beneficial to flatten the surgical hierarchy (what function is it serving)? We posit that differences between elective and emergency work, and prophylaxis and treatment, are important and will require further consideration to inform intervention design. Qualitative research addressing these questions may deliver important knowledge to bring about much needed change. However, it is important that the research is written up in a way that captures the richest insights, and is amenable to synthesis.

## Theoretical implications

The conceptual model makes visible the important influence of social and structural/environmental mediators on surgical APB in hospital settings, indicating that APB needs to be

reconceptualized as a multi-dimensional problem, consistent with previous research [68]. Interventions to improve surgical APB will need to take a systems approach rather than just targeting individuals [77]; and co-creation, with surgical teams, is likely to be important to ensure engagement, ownership [65] and ultimately sustainability, for example through the application of Participatory Action Research [78]; or a Change Laboratory [79].

Implementation scientists advocate the use of explicit theory when developing interventions [80,81]. In keeping with this, to further unfold the complexity of surgical APB and as the next step to informing intervention design, we propose cultural–historical activity theory (CHAT) as a potentially helpful framework. This approach is especially useful, allowing a paradigm shift in thinking about prescribing, moving from a focus on the internal workings of the individual (as with many behavioral change theories e.g. the COM-B model [82]) towards one encompassing their social and cultural aspects and surroundings [83]. Activity is understood as collective and object-oriented and challenges ('contradictions') are viewed as important triggers for learning and organizational change. CHAT is particularly well suited to the analysis of surgical APB in hospital settings as it embraces complexity and disturbances as an inherent feature of work processes [84,85].

## Strengths and limitations

The systematic search enabled us to locate suitable papers from across the surgical pathway and in a range of settings (e.g. adult and pediatric care; higher and lower income countries; private and state funded health-services) giving us a broad perspective on surgical APB. It also means the findings are more likely to resonate with our target audience [44]. The involvement of multiple reviewers (including those with extensive qualitative/meta-ethnography experience) during screening and analysis, and at regular meetings throughout the research, adds rigor [40,86] and demonstrates our engagement with the interpretive nature of the data. Furthermore, the involvement of a broad range of stakeholders, with first-hand surgical experience (healthcare workers and patients), adds credibility to the findings. Reflexivity was integrated throughout the research process e.g. guiding our decision to conduct regular reviewer meetings, and to involve stakeholders, enabling us to challenge our own understandings and to explore and test a range of possible analytic interpretations. Additionally, the use of meta-ethnography to address our research question is novel and has enabled us to develop a new conceptual model that provides insight into how we might improve surgical APB, moving the field forward by building on what was already known.

We identified five main challenges including: (1) the inherent difficulties in searching for qualitative literature [50] mean that we cannot be certain that all relevant studies were captured, and as such we cannot know if we missed a study that could have impacted our findings. However, strident efforts were made to find papers (see methods, authors' information and supporting information). (2) The meta-ethnographic approach relies on primary papers to provide 'data' and it is increasingly recognized that conceptually rich papers are pre-requisite [40]. However, much of the literature in this field is predominantly descriptive and atheoretical–perhaps in part due to the journals in which the papers are published–limiting the theoretical assumptions that can be inferred. Also, many papers do not appear to be crafted with an eye to being an antecedent to a synthesis. For example, key information is missing with regards the participants such as their specialty; and/or participant quotes are not contextualized with the participant's demographic information. (3) Our inclusion criteria specified 'peer-reviewed journal articles in English' meaning that we may have missed informative literature that has, as yet, not been published in English. Additionally, several of the included papers drew on the same dataset which some may argue could lead to the over-representation of certain concepts.

However, because meta-ethnography relies on the abstraction of ideas (not numerical data) this should not negatively impact the synthesis [87] and may enhance explanatory power through the incorporation of differing theoretical perspectives/lenses. For example, two Broom papers [28,57], with a shared data set, provided different insights and contributed to different overarching concepts (see Table 4) through the application of a differing methodological lens. (4) It is unclear how the order of reading and synthesizing the papers affects the findings or whether one method is better than another is [40,52]. We first read the papers chronologically but mitigated against a strong impact from this approach by revisiting included papers in different sequences, e.g. by author/research group; and by re-interrogating each concept's affiliation to an overarching concept (see methods). Lastly, (5) the meta-ethnography process is inherently interpretive (and still developing with multiple unanswered questions)–consequently different research teams will arrive at different syntheses and this work represents one version but not a single 'truth'.

## Conclusions

This study furthers understanding of how and why contextual factors act and interact to influence surgical antimicrobial prescribing behavior. Our synthesis posits that, social and structural mediators influence complex individual antimicrobial judgements skewing practice towards increased and unnecessary antimicrobial use resulting in avoidable patient harm, AMR and cost. It is clear that APB is complex, and that currently healthcare workers deploy antimicrobials across the surgical pathway as a safety net–to allay fears, reduce uncertainty and risk, and to mitigate against personal blame. Prescribing is unlikely to change until the social and structural mediators driving practice are addressed, and our conceptual model offers insights into how this may be approached.

This meta-ethnography highlights the urgent need for further qualitative research to explore the context for effective and sustainable quality improvement stewardship initiatives across surgical specialties. Next, we plan to explore the context for change focusing on the social and structural mediators of APB. For example, with surgical stakeholders we will further explore the role of hierarchy and how to secure surgical buy-in, which is likely to be key to eliciting change. This will allow us to better understand what solutions are likely to be acceptable and feasible for surgical teams, therefore enabling successful and sustainable adoption such that we can tip the balance towards more evidence-based antimicrobial use, to improve patient care.

## Supporting information

**S1 Checklist. PRISMA checklist.**
(DOCX)

**S1 File. Full literature search strategies.**
(DOCX)

## Acknowledgments

We would like to gratefully acknowledge the important contributions of: Prof Nicky Britten who contributed to the conception of the meta-ethnography and development of the protocol (Parker et al., 2020); Dr Emma Cockcroft who has supported the integration of public and patient involvement (PPI) throughout the research; and to the many patient representatives and healthcare workers who have generously shared their experiences and ideas to support the

research, particularly Dr Melissa Bennett; Dr Alex Burns; Mr Leon Farmer; Mr Odran Farrell; Ms Diana Frost; Dr Joel Prescott; Dr Harry Pugh; and Dr Alexis Taylor.

## Author Contributions

**Conceptualization:** Hazel Parker, Julia Frost, Karen Mattick.

**Data curation:** Hazel Parker, Sophie Robinson.

**Formal analysis:** Hazel Parker, Julia Frost, Jo Day, Karen Mattick.

**Funding acquisition:** Hazel Parker, Karen Mattick.

**Investigation:** Hazel Parker.

**Methodology:** Hazel Parker, Julia Frost, Jo Day, Sophie Robinson, Karen Mattick.

**Project administration:** Hazel Parker.

**Supervision:** Julia Frost, Jo Day, Karen Mattick.

**Visualization:** Hazel Parker, Rob Bethune, Anu Kajamaa, Kieran Hand, Karen Mattick.

**Writing – original draft:** Hazel Parker.

**Writing – review & editing:** Hazel Parker, Julia Frost, Jo Day, Rob Bethune, Anu Kajamaa, Kieran Hand, Sophie Robinson, Karen Mattick.

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
