## [Decision Letter · Decision Letter 0]

29 Mar 2022

PONE-D-21-40196Tipping the balance: a systematic review and meta-ethnography to unfold the complexity of surgical antimicrobial prescribing behaviour in hospital settingsPLOS ONE

Dear Dr. Parker,

Thank you for submitting your manuscript to PLOS ONE. After careful consideration, we feel that it has merit but does not fully meet PLOS ONE’s publication criteria as it currently stands. Therefore, we invite you to submit a revised version of the manuscript that addresses the points raised during the review process.

As both of the reviewers note, your manuscript provides a valuable and unique perspective on the antibiotic prescribing practices of the surgical specialties. The reviewers provide thoughtful feedback which I anticipate will strengthen the manuscript. Please respond to their comments and edit the manuscript accordingly. Also, I saw that you wrote to the PLoS One staff concerned about the timing of the review. My apologies for that. It turns out that it took some time to find suitable reviewers as folks are busy.

We look forward to receiving your revised manuscript.

Kind regards,

Seth Blumberg

Academic Editor

PLOS ONE

Journal Requirements:

Additional Editor Comments (if provided):

A few additional thoughts:

- Regarding the reviewer comment, ‘The introduction of inclusion criteria for key articles on lines 311-6 uses a “value” in somewhat of a judgmental frame’. Besides shifting the language, I would consider dropping the (*) designation in Table 3.

- While antibiotic prescribing practices can and should be improved, I think there has been noteworthy advances in the past couple decades (e.g. many hospitals have antibiotic stewardship programs and/or dedicated ID pharmacists that provide structural improvements to prescribing patterns). It might be useful to provide a bit of that context in the introduction.

- While framing the problem, it may help to explain a bit more about why excess antibiotics aren’t only harmful, but also not helpful (e.g. even though surgical site infections are common and problematic, prolonged prophylactic antibiotics typically don’t prevent them from occurring)

Reviewers' comments:

Reviewer's Responses to Questions

**Comments to the Author**

1. Is the manuscript technically sound, and do the data support the conclusions?

Reviewer #1: Yes

Reviewer #2: Yes

2. Has the statistical analysis been performed appropriately and rigorously? 

Reviewer #1: N/A

Reviewer #2: Yes

3. Have the authors made all data underlying the findings in their manuscript fully available?

Reviewer #1: Yes

Reviewer #2: Yes

4. Is the manuscript presented in an intelligible fashion and written in standard English?

Reviewer #1: Yes

Reviewer #2: Yes

5. Review Comments to the Author

Reviewer #1: Summary: Parker et al. investigate the critically important clinical problem of surgical antibiotic misuse and overuse, which has been challenging to solve due to the social and behavioral underlying complexities. The authors point out that structural and subjective factors drive unilateral and hierarchical decision-making by surgeons who often do not have the necessary expertise. Senior surgeons, as well as residents, understand the threat of antimicrobial resistance, but they are structurally motivated to reduce the risk of immediate severe infection over the distant and abstract threat of evolution of resistance.

While much research has focused on quantitative analyses of prescribing decisions and within-institution patterns of surgical antibiotic use, very little research has focused on qualitative assessments of prescribing behavior that includes cultural norms, hierarchies, among other self-assessments. Parker et al. conducted a “meta-ethnography,” which is a meta-analysis of the current field of knowledge that specifically focuses on qualitative studies. Antibiotic mis- and overuse is an especially urgent problem in surgical service because of the high infection risk and because surgeons have been reported to be less likely than other physicians to consult standard guidelines.

This is a thorough and important study that offers rich insights into the complex social system surrounding surgical antimicrobial prescription behavior. I suggest a few major (ish) changes and several minor changes to improve the message of the paper.

Major (ish):

The role of the including a stakeholder engagement stage in the analyses is not clear. Where, when, and how did these stakeholders enrich, critique, or contextualize meta-ethnographic findings? Are there instances where their input altered or contradicted the meta-ethnographic findings? If so, how is this justified? You also say on line 548 that they favored your conceptual model from Fig. 2, but it is not clear how that process occurred. Who were stakeholders and how were they recruited? A table that enumerates this participant group could be helpful.

It seems important to clarify the amount of overlap among the studies. The authors state that there are 10 unique authors among the 14 analyzed papers. How similar are the overlapping papers? Did they analyze the same datasets? If so, can we count them as totally separate studies? If this does not matter for this meta-ethnographic approach, that should be stated and justified.

Page 33, section about Line-of-argument synthesis (line 529):I do not understand line of argument synthesis as a method. I don’t expect most of your readers to know either, so I recommend giving a little more description of this component of your analysis.

If hierarchy is the main problem that feeds into the other components of the conceptual model, how do you get buy-ins from surgeons, who you report also generally have confidence in their own decisions, lack confidence in approved guidelines, and actively prioritize infection risk reduction over preventing evolution of resistance? You say that stewardship teams should cultivate better relationships with senior surgeons, but how do you get surgeons to participate in this?

On line 654, you hit what I perceive from your research as the real “rub,” which is that the hierarchy needs to be flattened, but you also recognize that this might not be possible. I agree that this could be the point of further qualitative (and quantitative) analysis.

Minor:

Just a quibble, but this phrase on p. 27, line 382-3: “…but a high plausibility of infection in the minds of the surgeons,” seems glib. There is a high risk of infection after surgery. It is one of the greatest contributors to hospital acquired infection, so it is not just in the surgeon’s mind. Supporting your argument, it may be that the fear of an infection and the resulting consequences drive surgeons to over-prescribe, but that risk is quite real.

Also, I’m curious to know why surgeons are more likely than other physicians to be distrustful of literature and guidelines, and why the perceive gaps in knowledge. Is this explained by some of the cultural and contextual themes you identified for prescribing behavior? This outlying perception seems like an important context in its own right.

Re-consider whether the long description of the co-authors’ backgrounds is necessary (p. 7, under “Theoretical perspective”), given this is already a fairly long paper.

The quote starting at line 576 is not elucidating. It’s obvious that the person is speaking metaphorically when they invoke “fairies,” but the metaphor does not lay bare what this person is actually saying. It’s just confusing.

Please check whether PLoS ONE requires American spelling. If not, just be sure UK spelling is consistently applied.

Some copy editing is required. Punctuation, particularly use of commas, hyphens, and semi-colons, is strange and incorrect throughout.

Reviewer #2: This is an important synthesis of qualitative and mixed-methods research to date on antibiotic prescribing in surgical specialties. In particular, the manuscript has the potential to bolster the representation of social and cultural mediators in research on antibiotic prescribing. A main area needing additional attention in the manuscript relates to the interdependencies in the themes. Two secondary areas include the descriptions of scale in the framing of the research problem and the introduction of inclusion criteria for key articles.

Hierarchy is a well-known theme in hospital settings. It is therefore fitting that hierarchy made it into your analysis as a theme. Yet, in relationship to the second and third theme, hierarchy is arguable conceived of as a pre-condition for fear driving action and deprioritization. In the sections on fear driving action and deprioritization, lines 384, 397, and 410 describe the role of surgeons, a topic much related to the theme of hierarchy. Please review this conceptual relationship and state the interdynamics among themes more clearly in the manuscript.

The description of scale in the introduction leaves the impetus for completing the study (a lack of inclusion of qualitative data in planning of antimicrobial stewardship) and the choice of methods for the study (a large scale, “meta” project) at odds. A more careful move from the local, in-depth scale of qualitative projects to the synthetic, broader scale of the analysis would be useful. Please explain the positionality of the study. The explanation of “cumulative insights” on pg. 8 could be useful earlier on in the manuscript.

The introduction of inclusion criteria for key articles on lines 311-6 uses a “value” in somewhat of a judgmental frame. While the end of the paragraph describes value in relationship to the research question the team generated, the earlier sentences do not invoke this rationale and simply label some articles’ value versus other articles. A suggestion here could be to shift the language from valuable/less valuable to assessments of relatedness (e.g., or centrality) to the research question/theme. In considering this shift, the articles’ utility as research is not question, just their relationship to the selected theme.

Finally, the sentence on lines 650-1 could use revising. How are these two clauses unrelated (i.e., on separate hands)? The sentence in the introduction referencing citation #9 also could use clarification on whether this is globally or within the UK? Review pgs. 3 and 4 for global vs. UK relevance.

6. PLOS authors have the option to publish the peer review history of their article (what does this mean?). If published, this will include your full peer review and any attached files.

Reviewer #1: No

Reviewer #2: No

---

## [Author Response · Author response to Decision Letter 0]

10 May 2022

Please see our cover letter where we have addressed each of the reviewer and editor comments in tables

---

## [Decision Letter · Decision Letter 1]

1 Jul 2022

Tipping the balance: a systematic review and meta-ethnography to unfold the complexity of surgical antimicrobial prescribing behavior in hospital settings

PONE-D-21-40196R1

Dear Dr. Parker,

We’re pleased to inform you that your manuscript has been judged scientifically suitable for publication and will be formally accepted for publication once it meets all outstanding technical requirements.

Kind regards,

Seth Blumberg

Academic Editor

PLOS ONE

Additional Editor Comments (optional):

Thank you for the thoughtful attention to reviewer comments. Congratulations on completing this insightful and compelling study!

Reviewers' comments:

Reviewer's Responses to Questions

**Comments to the Author**

1. If the authors have adequately addressed your comments raised in a previous round of review and you feel that this manuscript is now acceptable for publication, you may indicate that here to bypass the “Comments to the Author” section, enter your conflict of interest statement in the “Confidential to Editor” section, and submit your "Accept" recommendation.

Reviewer #1: All comments have been addressed

Reviewer #2: All comments have been addressed

2. Is the manuscript technically sound, and do the data support the conclusions?

Reviewer #1: Yes

Reviewer #2: Yes

3. Has the statistical analysis been performed appropriately and rigorously? 

Reviewer #1: Yes

Reviewer #2: Yes

4. Have the authors made all data underlying the findings in their manuscript fully available?

Reviewer #1: Yes

Reviewer #2: Yes

5. Is the manuscript presented in an intelligible fashion and written in standard English?

Reviewer #1: Yes

Reviewer #2: Yes

6. Review Comments to the Author

Reviewer #1: Thank you for your thoughtful responses to our comments. This is a very important paper and your updated manuscript will be an excellent guide for future meta-ethnographic work. I look forward to the future phases of this research.

Reviewer #2: Thank you for the opportunity to review this revised manuscript. The clear improvements include additional clarity on methodology and inclusion criteria. Removing the delineation of valuable papers has made the study much more readable.

The implications of this paper are important. Recommendations for change are given towards the end of the paper, with the suggestion being culture and social dynamics must change (lines 681-92). However, the authors also state that their overall goal is to further understand of antimicrobial prescribing. While many papers do begin with an impetus for understanding and end with a suggestion of change, I would urge the authors to consider the cycle of research that targets prescribers and prescribing habits in lieu of expanding a view towards systemic change. Beyond this comment, I do believe the manuscript to be much improved and close to ready for publication.

7. PLOS authors have the option to publish the peer review history of their article (what does this mean?). If published, this will include your full peer review and any attached files.

Reviewer #1: No

Reviewer #2: No

---

## [Editor Report · Acceptance letter]

11 Jul 2022

PONE-D-21-40196R1 

Tipping the balance: a systematic review and meta-ethnography to unfold the complexity of surgical antimicrobial prescribing behavior in hospital settings 

Dear Dr. Parker:

I'm pleased to inform you that your manuscript has been deemed suitable for publication in PLOS ONE. Congratulations! Your manuscript is now with our production department. 

Kind regards, 

on behalf of

Dr. Seth Blumberg 

Academic Editor

PLOS ONE